# Effects of a Mindfulness and Physical Activity Programme on Anxiety, Depression and Stress Levels in People with Mental Health Problems in a Prison: A Controlled Study

**DOI:** 10.3390/healthcare11040555

**Published:** 2023-02-13

**Authors:** Jose Gallego, Adolfo J. Cangas, Israel Mañas, Jose M. Aguilar-Parra, Álvaro I. Langer, Noelia Navarro, Maria-Jesus Lirola

**Affiliations:** 1Department of Education, Faculty of Education Sciences, University of Almería, 04120 Almería, Spain; 2Department of Psychology, HUM-760 Research Team, Health Research Centre, University of Almería, 04120 Almería, Spain; 3Department of Psychology, HUM-878 Research Team, Health Research Centre, University of Almería, 04120 Almería, Spain; 4Institute of Psychological Studies, Faculty of Medicine, Austral University of Chile, Valdivia 5090000, Chile; 5Faculty of Psychology and Humanities, Universidad San Sebastián, Valdivia 5111642, Chile; 6Millennium Nucleus to Improve the Mental Health of Adolescents and Youths, Imhay, Santiago 8380455, Chile

**Keywords:** mindfulness, severe mental disorder, well-being, exercise

## Abstract

Recent studies in the general population have shown an inverse relationship between mindfulness and symptoms of anxiety and depression, as well as the benefits of physical activity on these symptoms. These relationships have not yet been studied in a population with severe mental disorder (SMD) in prison, where symptoms of anxiety and depression and impulsive behaviours have a high incidence. A controlled study was developed to assess the benefits of a mindfulness-based protocol whereby elements of Acceptance and Commitment Therapy were evaluated and compared to an adapted sport program. Twenty-two inmates from the “El Acebuche” prison aged between 23 and 58 years old participated in this study, which included a pre-, post-, and follow-up; the majority of participants had SMD and were distributed in both conditions. The DASS-21 was obtained for its evaluation. The results of the Mann–Whitney U test for independent samples indicated a significant reduction in the levels of stress and depression in the mindfulness intervention group compared to the control group in which no significant changes were observed, providing evidence on the effect of this practice in prison contexts.

## 1. Introduction

Several studies have shown the effectiveness of mindfulness in treating different psychological problems such as anxiety, depression and somatisation [1], as well as in severe mental disorders (SMD) such as psychosis or schizophrenia [2,3,4,5]. On the other hand, there is evidence that sports and physical activity can improve depression, stress and anxiety in prison inmates, which are problems manifested with a high frequency [6,7,8].

Nevertheless, despite the many existing studies in the general population with these interventions, their application in prison contexts has not been as common. However, existing studies have found that both mindfulness and sports can achieve improvements in inmates’ stress, anxiety and depression [9]. However, despite the benefits of mindfulness, there are no studies where it has been applied to people who, on the one hand, have a SMD and, on the other hand, are inmates in prisons. In this regard, it is worth noting that the deprivation of liberty is reported as a highly stressful experience [10], hindering the satisfaction of needs, privacy, security and the fulfilment of goals, while at the same time distancing them from support networks [10,11].

Furthermore, it has been common to methodologically compare the results of a mindfulness group with a waiting control group, but not so much with other effective interventions. For example, the effects of physical activity on anxiety and depression problems are well documented, with results in some studies equivalent to pharmacological treatment and superior to the placebo effect [12]. Some research points out precisely that the results of mindfulness are not as high if they are not compared to an active group [13].

To date, what has not been carried out is a joint comparison of both interventions, that is, a test of whether mindfulness or a physical activity programme can have a superior effect on any variable. This was precisely the aim of the present study, to compare the effect of both interventions on anxiety, depression and stress in a group of inmates belonging to the Framework Programme for the Integral Assistance to the Mentally Ill in Prisons (PIAMI).

## 2. Method

### 2.1. Design

This randomized, controlled study was conducted at “El Acebuche” Prison Centre. Eligible patients were randomly assigned to the different groups (i.e., the mindfulness group and sport group). Study site visits occurred at the prison twice a week with 1 h working session in each group during eight consecutive weeks for a total of 11 sessions. All the participants were selected by the prison board of directors, and the participants provided written informed consent before participating in the study.

### 2.2. Participants

An incidental sampling was carried out by the medical team, and the sample consisted of 23 inmates belonging to the El Acebuche Penitentiary Centre in Almería and the Framework Programme for the Integral Assistance to the Mentally Ill in Penitentiary Centres (PIAMI), of whom 60.87% (n = 14) suffered from schizophrenia, 17.39% (n = 4) from affective disorders and 21.74% (n = 5) from personality disorders. Additionally, 43.48% regularly or sporadically used various substances, the most common substance being hashish at 30.43%, followed by cocaine at 13.04%. The inclusion criteria for both groups were as follows:Belonging to the PIAMI programme of the *El Acebuche* prison in Almería.Availability of the inmate throughout the study, i.e., not being close to release.Not presenting behavioural reports for inappropriate behaviour (internal prison measure for the enjoyment of unofficial activities), i.e., being among the inmates with the best behaviour.Understanding of the language.

Participants ranged in age from 23 to 58 years (*M_age_* = 42.86; *SD* = 9.05). All patients were male. Table 1 and Table 2 provide an individual description of these variables for each participant. The participants were randomly assigned to either the mindfulness or the sport group.

### 2.3. Instruments

The Depression, Anxiety and Stress Scale (DASS-21) is a shortened version of the DASS [14], measuring negative emotional states of depression, anxiety and stress. It contains 21 items with four Likert-type response alternatives referring to the participants’ experiences during the past month, ranging from never happened to almost always happened to me. The respondents were asked to answer according to how often they had experienced each state during the month prior to the time of the survey. The psychometric properties reported by Henry and Crawford [15] explained 49% of the variance, and a Cronbach’s alpha coefficient of 0.93 was found for the total instrument in a study conducted with 1700 adult volunteers from the open population in the British Union; additionally, good reliability indices were also recently reported with a Cronbach’s alpha of 0.86 using a nonclinical sample in a Spanish-speaking country [16].

An attendance register was used for both groups since among the criteria for belonging to the group, an inmate who missed more than two sessions would be excluded from the sample under study.

### 2.4. Procedure

Once the relevant permits were obtained, the prison medical team selected inmates who were part of the PIAMI programme (a nonprobabilistic sample). The participants were randomly divided into two groups, one to participate in the mindfulness group and the other in the sport group. All the participants were adequately informed about the programme and signed an informed consent form. Figure 1 visually shows the process followed in this research.

Two of the participants who started the programme did not complete it due to a change in the prison and an accumulation of behavioural parts. After a first briefing session, all the inmates were assessed with the DASS-21 scale. During the following seven weeks, the intervention was carried out in both groups in two weekly sessions of two hours duration for a total of 11 sessions. After the end of the intervention, the post-test evaluation was completed, and two and six months later, the follow-up evaluation was conducted. The exercise sessions were provided by a professional with a university degree in Physical Activity and Sport Sciences with more than five years of experience, and for the mindfulness group, the responsible guide was a psychologist with more than five years of experience in the development of sessions. Data collection was conducted in the presence of two of the researchers of this study. The data were processed with the SPSS 24 statistical package. All the principles of the Singapore Declaration were respected.

### 2.5. Sport Group

The venue for the sporting activities was the outside courtyard of the prison, a large place where inmates, officials, nurses and doctors were led by a professional in Physical Education and Sport with extensive experience in working with individuals with mental illnesses. The sessions were selected from the protocol carried out by Mullor et al. [17] for people with severe mental disorder problems and always encouraged values such as respect, group spirit and cooperation, as well as developing sports activities such as brisk walking, stretching, running and team sports. Two sessions of approximately one hour per week were carried out over 8 consecutive weeks. The participants were also asked to continue these practices autonomously outside the supervised sessions, which they were asked to record in a register with the aim of making sport a daily habit.

### 2.6. Mindfulness Group

Initially, a weekly session was established, guided by a mindfulness expert with more than five years of experience in the application of mindfulness. The sessions were structured in three parts. First, they began with a summary of the previous session, observations and the handing in of records. Then, formal meditation was practised for ten minutes in the first session, fifteen minutes in the second and twenty minutes in the remaining sessions under the following instructions: closing the eyes, sitting comfortably in a relaxed position, becoming aware of the body, focusing on the sensations of the breath, returning attention to the breath if it is lost and maintaining an attitude of acceptance and curiosity about the experience of the present moment. Finally, we worked with different exercises, such as body scanning [18], metaphors, paradoxes and “here and now” exposure exercises based on relational learning from Acceptance and Commitment Therapy [19,20]. Once the participants became proficient in these practices (from the fourth session onwards), a second weekly session was added in which they worked on body awareness through an audio of the mindfulness-based stress reduction (MBSR) programme, a variant composed of three main techniques such as the mindfulness of seated breathing, body-scanning and yoga [18].

All the sessions were held in the prison assembly hall, a spacious and soundproof room. Individual homework was requested outside the sessions, which had to be recorded in a register.

The duration of the procedure was eight weeks, following the duration criteria of previous interventions where improvements were found in that time frame [21,22]. At the end of the procedure for both groups, the DASS-21 was administered again (the post-treatment measure), as well as at two and six months after the end of the intervention (the follow-up).

### 2.7. Data Analysis

In order to assess whether there were differences between the scores of the different questionnaires in the pre- and post-follow-up measures at 2 and 6 months, the Friedman test for related samples was applied for both the mindfulness group and the sport group. This is the nonparametric alternative for repeated measures as is the case here. In addition, the Mann–Whitney U test for independent samples was applied to compare whether there were differences between the two intervention groups in relation to each phase of the study and in each subscale. In addition, the Wilcoxon rank sum test was used to determine the existence of differences between the four measures developed in each group by means of nonparametric multiple comparisons. The effect size was used to quantify the size of the difference, in this case, by means of Rosenthal’s r statistic [23], which corresponds to the tests performed.

## 3. Results

The attendance rate of the mindfulness group (N = 11) was 93.28% and of the sport group (N = 11) was 93.68%. The mean scores of the depression variable indicated a moderate level of depression in the initial assessment in both groups, which was maintained in the postassessment. However, at the two-month follow-up, there was a significant decrease in both groups to normal levels, and it finally became mild at the six-month follow-up. Although these changes were not statistically significant, possibly due to the small sample size, when corrected with the Wilcoxon rank sum test, statistically significant differences were observed in the sport group between the post- and two-month follow-up and in the mindfulness group between the pre- and two-month follow-up. Furthermore, if we look at the effect size, we can see that there were changes, moderate in the case of the sport intervention and moderate and high in the case of mindfulness, following these interventions as shown in Table 3.

In terms of anxiety, the mean score for both groups remained at moderate levels at the pre- and postassessment. However, at the two-month follow-up, the levels dropped to mild, almost normal, although at the six-month follow-up they rose again to moderate anxiety levels. However, as with depression, these differences were not statistically significant; only between the post-test and the two-month follow-up in the case of the sport group were significant differences found with a strong intervention effect. In the case of the mindfulness group, differences were found between the post-test and the two-month follow-up with a moderate effect size.

The most significant changes were in the stress levels, where the mindfulness group started at a medium stress range and reduced their stress in the post-test evaluation to a normal level, which was maintained at the two- and six-month follow-up. The sport group started at a normal stress range, increased in the post-test to medium levels and decreased again at the two- and six-month follow-up to normal levels. In the case of mindfulness, the effect size of the intervention (*r*) caused strong changes in stress levels, while in the case of the sport programme, it reported moderate changes.

Differences between the two intervention groups were tested in relation to each phase of the study and in each subscale, as shown in Table 4 and Table 5, and no statistically significant differences were found between the groups in any of the subscales or at any of the times evaluated. The effect size was moderate only in the post-test stress measurement, which suggests that at that time of the study, the differences between the two groups were somewhat more prominent.

## 4. Discussion

The purpose of this study was to compare the effectiveness of a novel intervention such as mindfulness with another more consolidated and practised intervention such as a physical activity programme in a prison setting for people with mental disorders. In this sense, the first aspect that is relevant to highlight is the high percentage of attendance in both groups; these data suggest that the inmates were highly interested in attending the sessions, taking into account their voluntary nature and the fact that their attendance did not have any repercussions in the prison regime. It is also noteworthy that this is the first study to apply a mindfulness-based intervention that specifically incorporated a high percentage of people with SMD (59.09% of inmates with schizophrenia) in a prison context. In this way, the present study provides evidence of the applicability of mindfulness in SMD in prison contexts.

The results showed that there were no major differences in scores between the two intervention groups. These results are in line with the meta-analysis conducted by Nidich et al. [13], which showed that, in prisons, the effect size of psychological interventions (including mindfulness-based interventions) that reduce symptoms of mental health problems is small when compared to active controls, as was found in the present study.

It is also noteworthy that both conditions were able to reduce depressive symptomatology to normal levels at the two-month follow-up after the end of the intervention. Similarly, a recent study showed that both a mindfulness intervention and a communication skills intervention reduced anxiety symptoms in men in prison who were in drug treatment [24]. Additionally, scholars have found a reduction in depressive symptomatology and improvement in recovery in people with schizophrenia [25], as was found in this investigation, where the majority of them presented with a severe mental pathology, especially schizophrenia.

An additional difference between the two groups that reached significance was the stress subscale, where statistically significant differences in favour of the mindfulness group were observed in the post-test. However, if we take into account the intragroup results, the mindfulness group reduced its stress levels to normal ranges and maintained them two months later, compared to the adapted sport group. Furthermore, after the follow-up assessment, both groups showed a reduction in their mean levels of depression, anxiety and stress to normal levels, but this was only significant for the mindfulness group for the stress and depression variables. Thus, the results confirmed the pre- and postchanges in anxiety or stress symptomatology that was observed in other studies that employed mindfulness in both women and men in prison [26,27].

This result could be explained by the process that is acquired with mindfulness practice, namely regulating attention and being aware of the present moment, which can lead to behavioural changes and a reduction in reactivity and therefore stress. On the other hand, the programme used, MBSR, focuses specifically on stress reduction and emotional self-regulation, as well as improving cognitive development, relaxation and acceptance of the self [28]; thus, the results are in line with expectations. All this would point to a favourable attitude towards learning and the generalisation of its use.

The limitations of the study include the type of nonrandom, incidental sampling, which may have generated extraneous or contaminating variables beyond our control, such as the selection of inmates who showed the best preintervention behaviour. Voluntary participation may have also been under the control of obtaining benefits other than those obtained in the study, which may have led to some participants not really becoming involved. Individual preferences were also not taken into account when allocating the groups, nor was it possible to isolate the effect of the instructor and the group itself. The same applies to the social desirability mentioned above, a phenomenon of even greater relevance in this type of context. It should not be forgotten that the sample was not homogeneous in terms of diagnosis and was very small and only included males, which makes it difficult to generalise the data obtained here.

Nevertheless, and despite the limitations mentioned above, the data shown here could be an excellent starting point for future research into the development and application of group protocols in the prison context, as well as into the capacity of mindfulness in these contexts and with the mentally disturbed population to establish a conscious relationship with emotions, thoughts and sensations.

As a final conclusion, this study is of interest for the improvement and implementation of programmes based on mindfulness and physical exercise, and both have been shown to have a positive influence on the improvement of symptoms of depression, stress and anxiety in this research and in previous studies. In a restricted context and with specific particularities such as a penitentiary centre, being able to count on practices that can reduce these maladaptive states is of great interest. However, these findings can also be extended to other contexts in society where stress and anxiety levels seem to be increasingly high.

The growth of these psychological states over the past two decades in most Organisation for Economic Co-operation and Development (OECD) countries will translate into higher healthcare costs in the future [29]. The presence of strategies to increase social well-being and reduce the symptoms of maladaptive psychological states has led the World Health Organization [30] to promote practices such as those studied in this intervention (i.e., exercise and mindfulness). Thus, one of its objectives, to raise public awareness of the health significance of the problem of mental health, is now closer, as the media in modern society is a fundamental channel for obtaining information and changing citizens’ behaviours. Therefore, intervention projects such as those carried out in this research would lead to the continuation of the addition of political–practical lines to improve the quality of social welfare.

## Figures and Tables

**Figure 1 healthcare-11-00555-f001:**
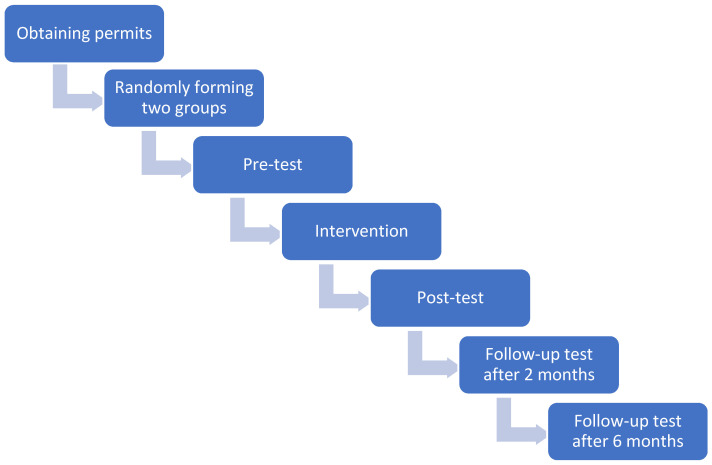
Steps followed in the investigation.

**Table 1 healthcare-11-00555-t001:** Sociodemographic characteristics of each participant in the mindfulness group.

Consume	Diagnosis	Age
Sporadic hashish	Schizophrenia	47
Hashish	Paranoid Schizophrenia	42
Alcohol	Schizophrenia	39
Hashish, cocaine sporadically	Narcissistic Disorder and Suicide Prevention	24
No consumption	Bipolar Disorder	53
No consumption	Bipolar Disorder	38
No consumption	Schizophrenia	51
No consumption	Schizophrenia	54
Hashish	Major Depressive Disorder and Antisocial Personality Disorder	52
No consumption	Delusional Disorder	46
No consumption	Schizophrenia and Pigmentary Glaucoma	39

Note. Informed consumption. Diagnosis according to DSM-V (APA, 2013).

**Table 2 healthcare-11-00555-t002:** Sociodemographic characteristics of each participant in the sport group.

Consume	Diagnosis	Age
No consumption	Personality disorder not specified	55
No consumption	Schizophrenia	41
Hashish, heroin, cocaine	Schizophrenia	34
No consumption	Schizophrenia	44
No consumption	Depression	45
Hashish, heroin, cocaine	Schizophrenia	40
Ecstasy, hashish, alcohol	Schizophrenia	20
Cocaine	Schizophrenia	38
Cocaine	Schizophrenia	39
No consumption	Schizophrenia	58
No consumption	Personality disorder not specified	41

Note. Informed consumption. Diagnosis according to DSM-V (APA, 2013).

**Table 3 healthcare-11-00555-t003:** Friedman test for repeated measures in the DASS 21 in the sport and mindfulness groups with respect to the study variables.

	Friedman				Wilcoxon				
			*M*	Sig.	*X* ^2^		Sig.	*Z*	*r*
**Depression**	**Sport**	**Pretest**	15.82	0.309	3.594	**Pre-post**	0.959	−0.051	−0.011
		**Post-test**	14.55			**Pre-segto2**	0.066	−1.837	−0.392
		**Segto.2**	7.18			**Pre-segto6**	0.262	−1.122	−0.239
		**Segto.6**	12.00			**Post-segto2**	0.036	−2.092	−0.446
						**Post-segto6**	0.722	−0.356	−0.076
						**Segto-segto6**	0.130	−1.512	−0.322
	**Mindfulness**	**Pretest**	16.36	0.100	6.241	**Pre-post**	0.474	−0.715	−0.152
		**Post-test**	14.00			**Pre-segto2**	0.017	−2.383	−0.533
		**Segto.2**	7.11			**Pre-segto6**	0.208	−1.260	−0.282
		**Segto.6**	10.89			**Post-segto2**	0.080	−1.752	−0.392
						**Post-segto6**	0.593	−0.534	−0.119
						**Segto-segto6**	0.084	−1.725	−0.407
**Anxiety**	**Sport**	**Pretest**	10.73	0.067	7.160	**Pre-post**	0.411	−0.821	−0.175
		**Post-test**	14.36			**Pre-segto2**	0.229	−1.202	−0.256
		**Segto.2**	7.64			**Pre-segto6**	0.246	−1.159	−0.247
		**Segto.6**	13.09			**Post-segto2**	0.013	−2.492	−0.531
						**Post-segto6**	0.959	−0.051	−0.011
						**Segto-segto6**	0.008	−2.670	−0.569
	**Mindfulness**	**Pretest**	13.78	0.230	4.310	**Pre-post**	0.799	−0.254	−0.054
		**Post-test**	12.67			**Pre-segto2**	0.103	−1.628	−0.364
		**Segto.2**	8.00			**Pre-segto6**	0.953	−0.059	−0.013
		**Segto.6**	11.89			**Post-segto2**	0.036	−2.094	−0.468
						**Post-segto6**	0.943	−0.071	−0.016
						**Segto-segto6**	0.066	−1.840	−0.434
**Stress**	**Sport**	**Pretest**	11.82	0.140	5.477	**Pre-post**	0.202	−1.276	−0.272
		**Post-test**	17.45			**Pre-segto2**	0.284	−1.071	−0.228
		**Segto.2**	8.64			**Pre-segto6**	0.372	−0.892	−0.190
		**Segto.6**	13.18			**Post-segto2**	0.026	−2.225	−0.474
						**Post-segto6**	0.477	−0.711	−0.152
						**Segto-segto6**	0.099	−1.647	−0.351
	**Mindfulness**	**Pretest**	16.44	0.032	8.793	**Pre-post**	0.009	−2.609	−0.556
		**Post-test**	9.33			**Pre-segto2**	0.011	−2.533	−0.567
		**Segto.2**	7.78			**Pre-segto6**	0.109	−1.602	−0.358
		**Segto.6**	10.56			**Post-segto2**	0.675	−0.419	−0.094
						**Post-segto6**	0.514	−0.653	−0.146
						**Segto-segto6**	0.261	−1.123	−0.265

Note. Pre = measurement before the intervention; post = measurement after the intervention; segto2 = measurement two months after the intervention; segto6 = measurement six months after the intervention; *M* = mean; Sig. = level of significance (*p* < 0.05); *X*^2^ = chi squared; *Z* = *Z* test; *r* = effect size.

**Table 4 healthcare-11-00555-t004:** Mann–Whitney U test to verify the differences between the sport group and the mindfulness group with respect to the study variables at pre- and post-test.

	Depression Pretest	Anxiety Pretest	Stress Pretest	Total Pretest	Depression Post-Test	Anxiety Post-Test	Stress Post-Test	Total Post-Test
**U Mann–Whitney**	59.50	56.00	41.00	56.50	60.00	47.00	36.00	50.50
**W from Wilcoxon**	125.50	122.00	107.00	122.50	126.00	113.00	102.00	116.50
** *Z* **	−0.066	−0.298	−1.286	−0.263	−0.033	−0.894	−0.614	−0.657
** *p* **	0.947	0.766	0.199	0.792	0.974	0.372	0.106	0.511
** *r* **	−0.014	−0.063	−0.274	−0.056	−0.007	−0.190	−0.344	−0.140

Note. *p* = level of significance (*p* < 0.05); *Z* = *Z*-test; *r* = effect size.

**Table 5 healthcare-11-00555-t005:** Mann–Whitney U test for differences between the sport group and the mindfulness group with respect to the study variables at 2- and 6-month follow-up.

	Depression Segto.2	Anxiety Segto.2	Stress Segto.2	Total Segto.2	Depression Segto.6	Anxiety Segto.6	Stress Segto.6	Total Segto.6
**U Mann–Whitney**	47.00	49.50	45.50	48.00	41.00	35.50	33.50	34.50
**W de Wilcoxon**	92.00	94.50	90.50	93.00	86.00	80.50	78.50	79.50
** *Z* **	−0.192	0.000	−0.305	−0.114	−0.655	−1.085	−0.236	−0.142
** *p* **	0.848	0.000	0.760	0.909	0.513	0.278	0.216	0.253
** *r* **	−0.042	0.000	−0.068	−0.025	−0.146	−0.242	−0.276	−0.255

Note: segto2 = measured two months after the intervention; segto6 = measured six months after the intervention; *p* = level of significance (*p* < 0.05); *Z* = *Z*-test; *r* = effect size.

## Data Availability

The original contributions presented in this study are included in this article, further inquiries can be directed to the corresponding author.

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
