# Peer review of "Effects of a Mindfulness and Physical Activity Programme on Anxiety, Depression and Stress Levels in People with Mental Health Problems in a Prison: A Controlled Study"

_healthcare, 2023, doi:10.3390/healthcare11040555_

Round 1

Reviewer 1 Report

1.   1.       Please add study type to abstract and methods part.

2.       Depression, Anxiety and Stress Scale please add reliability details.

3.       How many researchers collect the data? How many of them do intervention? Please give information about researcher background about yoga (certificated or not)

4.       Is this program reliable and valid? How many minutes did you do this procedure? How many weeks? How did you decide?

5.       Please add a diagram or figure for prodedure. It helps to understand study easily from a figure.

6.       I suggest that the authors rewrite the Discussion section. I'd like the results and discussion to be a bit stronger.  So, what makes your information new and different?  What does this information do to improve people with mental health problems in a prison? Most of the discussion paragraphs simply report the findings and then narrate different findings from related studies. What is missing in the discussion section is what these findings mean.

7.       I think the authors must make a more forceful statement in their conclusion.

Reviewer 2 Report

The reviewed article presents a clear objective and a relevant theme such as the effectiveness of intervention programs to address mental health and especially in a sample as vulnerable as people deprived of liberty.

It is precisely the selection of the sample, the type of sampling used and the assignment of groups, which, in my opinion, present the most deficiencies.

I share my comments and suggestions

Effects of a mindfulness and physical activity programme on 2 anxiety, depression and stress levels in people with mental 3 health problems in a prison: A controlled study

The reviewed article presents a clear objective and a relevant theme such as the effectiveness of intervention programs to address mental health and especially in a sample as vulnerable as people deprived of liberty.

It is precisely the selection of the sample, the type of sampling used and the assignment of groups, which, in my opinion, present the most deficiencies.

In the description of the sample, the count gives 21 participants, not 22. Almost 60% of the sample suffers from schizophrenia, in relation to the other mental disorders. One of the groups presents more varied pathologies, while in the other only 2 cases out of 11 do not present schizophrenia. When there is such a disparate percentage in this case between pathologies, the most convenient type of sampling to use could be proportional stratified random, respecting the proportion in each group, in this case of individuals with schizophrenia and that this difference in the assignment of the groups of not being contemplated can generate a bias or become a strange variable. Clearly this could have been a limitation, as expressed by the authors in the discussion section.

In the case of the analysis of the results and the discussion of the same, I consider that we must be even more cautious, when talking about reduction of symptoms or reaching normal levels of the same, then to the interventions, since we are talking about individuals with a diagnosis of severe mental pathology and especially schizophrenia.

We could even think that the group treated through the mindfulness program were the ones that were mostly diagnosed with schizophrenia, which could indicate that this intervention could be more effective in patients with this pathology, and not in all mental illnesses.

Finally, I am concerned about the voluntariness of these individuals, who are deprived of their liberty, have a mental illness and were previously selected by the medical team, when choosing to participate.

There really is room for refusal to participate in a situation like this. Hence, the look at exhaustive care or a greater description of the ethical aspects taken into account for this study.

Reviewer 3 Report

After a carefully revision, I have the follow suggestions. It is necessary to modify table 1, because is too large and it go off the page, thus, it could be a better option to separate the results in two tables. In addition, it is convenient to improve the explanation of the results included in each section, to do it, it is necessary to include their level of significance and explain the abbreviations used, these will be the results clearer and accurate all the tables.

For example: in table 1, third column of the Friedman test results, the column header says “Sig.”, what does this mean?, and in column 5 the result of 3.594 is written, what does it mean?, It does not correspond to the column header, this should be reviewed when data of the Wilcoxon and Mann Whitney U tests and effect size are showed, because there are missing information.

It is convenient to add a table with the descriptive data for each subscale and dividing by group.

Round 2

Reviewer 1 Report

Dear authors thank you very much for your revised version of the article.